# The New Challenge in Pediatric Liver Transplantation: Chronic Antibody-Mediated Rejection

**DOI:** 10.3390/jcm11164834

**Published:** 2022-08-18

**Authors:** Elena Yukie Uebayashi, Hideaki Okajima, Miki Yamamoto, Eri Ogawa, Tatsuya Okamoto, Hironori Haga, Etsurou Hatano

**Affiliations:** 1Department of Pediatric Surgery, Kyoto University Hospital, Kyoto 606-8507, Japan; 2Department of Surgery, Kyoto University Graduate School of Medicine, Kyoto 606-8507, Japan; 3Department of Pediatric Surgery, Kanazawa Medical University Hospital, Kanazawa 920-0293, Japan; 4Department of Diagnostic Pathology, Kyoto University Hospital, Kyoto 606-8507, Japan

**Keywords:** pediatric liver transplantation, chronic antibody-mediated rejection, chronic rejection, humoral rejection

## Abstract

Antibody-mediated rejection (AMR) of liver allograft transplantation was considered as anecdotal for many decades. However recently, AMR has gained clinical awareness as a potential cause of chronic liver injury, leading to liver allograft fibrosis and eventual graft failure. (1) Methods: Literature on chronic AMR (cAMR) in pediatric post-liver transplant patients was reviewed for epidemiologic data, physiopathology, diagnosis, and treatment approaches. (2) Results: Accurate incidence of cAMR in pediatric liver transplantation remains unknown. Diagnostic criteria of cAMR were suggested by the Banff Working Group in 2016 and are based on standardized histopathological findings, C4d staining pattern, associated with the presence of donor-specific antibodies (DSA). Physio-pathological mechanisms are not clear for the technically difficult-to-obtain animal models reproducing cAMR. Treatment protocols are not established, being limited to case reports and case series, based on experience in ABO incompatible transplantation and kidney transplantation. Immunosuppression compliance with adequate dose adjustment may prevent cAMR. Conversion of Cyclosporine to Tacrolimus may improve pathological findings if treated in early phase. The association of steroids, Mycophenolate Mofetil (MMF) and mTOR inhibitors have shown some synergistic effects. Second-line treatments such as intravenous immunoglobulin (IVIG) and plasma exchange may decrease antibody titers based on ABO incompatible transplant protocols. The use of anti-CD20 (Rituximab) and proteasome inhibitors (Bortezomib) is controversial due to the lack of qualified studies. Therefore, multicenter randomized trials are needed to establish the best therapeutic strategy. In refractory cases, re-transplantation is the only treatment for allograft failure. (3) Conclusions: This literature review collects recent clinical, histopathological, and therapeutical advances of cAMR in liver allograft transplantation of pediatric patients. There are many physio-pathological aspects of cAMR to be clarified. Further efforts with multicenter prospective protocols to manage patients with cAMR are needed to improve its outcome.

## 1. Introduction

Liver transplantation has become the mainstay of treatment for end-stage liver disease with improved outcomes in recent years. Technical advances have corroborated this improvement, and in Japan, pediatric post-liver transplant graft survival has reached 88.9% in the first year, 82.2% in the tenth year, 77% in the 20th year, and 75.4% in the 30th year [1]. Therefore, to improve the management of long-term complications, such as chronic antibody-mediated rejection (cAMR), becomes the next challenge to maintain a good quality of life in liver transplanted patients.

The liver has been considered “immune-privileged” with anergy for several antigens and a high rate of immune tolerance. AMR incidence in liver transplantation, estimated as 1%, seems to be much lower than in heart (10–30%) or kidney transplantation (20–50%) [2,3]. However, in the last decade, several groups have reported the association of donor-specific antibodies (DSA) with chronic rejection [4] and increased rates of liver fibrosis in apparently stable long-term liver-transplanted pediatric patients, suggesting the influence of AMR in liver allografts [5,6,7,8,9,10]. Here, we performed a descriptive review of the recent literature on liver cAMR in pediatric patients to clarify clinical and physio-pathological aspects and discuss the best management of cAMR.

## 2. Materials and Methods

### 2.1. Search Strategy

This descriptive review is based on the literature research of NCBI PubMed database to find relevant studies related to pediatric liver transplantation (LT) and chronic antibody-mediated rejection (cAMR). Full articles were accessed and reviewed. The following keywords were used in different combinations to reach the greatest number of articles: “pediatric”, “liver transplantation”, “chronic antibody-mediated rejection”, “humoral rejection”, “chronic rejection”.

### 2.2. Inclusion and Exclusion Criteria

Relevant studies of chronic antibody-mediated rejection in pediatric liver transplantation were selected and included in this descriptive review. Articles published in languages other than English were excluded.

## 3. Results

### 3.1. Definition of Chronic Antibody-Mediated Rejection (cAMR)

Until recently, antibody-mediated rejection was overlooked in liver transplantation. However, in the last decade, many groups have reported the influence of humoral immunity in liver transplantation [4,5,6,7,8,9,10,11,12,13,14,15,16,17,18]. Therefore in 2016, the Banff Working Group introduced the concept of antibody-mediated rejection for liver transplantation, suggesting that probable chronic AMR should include these four criteria (Table 1) [2,3,7,19]:Compatible histology (both required): (a) unexplained mononuclear portal and/or perivenular inflammation with interface and/or perivenular necro inflammatory activity, and (b) moderate portal/periportal, sinusoidal and/or perivenular fibrosis.Positivity for DSAs within 3 months of biopsy.Focal C4d positivity (>10% portal tract microvascular endothelia).Reasonable exclusion of other liver insults that may cause a similar pattern of injury.

Possible cAMR present similar findings, but C4d is minimal or absent.

According to this criterium, the diagnosis of cAMR is fulfilled only if the physician suspects, actively performing C4d staining, and checking patient’s DSA levels.

### 3.2. Prevalence

The true prevalence of cAMR is unknown, for most patients remain undiagnosed, with normal liver enzymes for long time, despite chronic allograft injury. It may be present in around 8–15% of the patients who present persistent or de novo DSA, specially targeting HLA class II [2,10]. Most patients are diagnosed as cAMR in protocol biopsies [4,5,6,7,8,9,10,11,13,14,15,19,20,21], tolerance-inducing weaning protocols [3,5,8,14,15,22], as well as biopsies performed in inexorable immunosuppression cessation secondary to infections (Epstein-Barr Virus, EBV; Cytomegalovirus, CMV), malignancies (post-transplant lymphoproliferative disease, PTLD) or non-compliance [2,3,4,5,6,8,9,10,12,13,14,19,23,24].

### 3.3. Physio-Pathology

The pediatric population has fewer recurrences after LT (such as viral hepatitis) compared to the adult population, leading to fewer biases to analyze the relationship between DSA and liver fibrosis in order to study cAMR. Most of the data on chronic AMR are based on long-term followed-up patients with protocol biopsies associated with DSA and C4d evaluation in the pediatric population.

As demonstrated in the Figure 1, organ transplantation from allogeneic donor may cause activation of T-lymphocytes leading to the T-cell mediated rejection (TCMR), as well as activation of B-lymphocytes leading to the production of donor specific antibodies (DSA) against the human-leucocyte antigen (HLA) epitopes present in the graft, leading to AMR. HLA class I are expressed on all cells in the liver, while class II are expressed in dendritic cells (DC), Kupffer cells (KC) and weakly in portal and sinusoidal endothelial cells (EC). The DSA is preformed or can emerge at any time after transplantation (de novo DSA) being related to insufficient immunosuppression for non-compliance, IS withdrawal due to viral infection, PTLD or tolerance-induction protocol [2,23,25].

AMR involves activation of the classical pathway of complement cascade by DSA, resulting in endothelial injury. Several effector mechanisms are activated: natural killer (NK) cells produce and release interferon-γ (IFN-γ), tumor necrosis factor (TNF), and granulocyte-macrophage colony-stimulating factor (CSF2), acting on monocytes which release TNF, interleukin-1 (IL-1), interleukin-6 (IL-6), damaging and activating endothelial-cell targets and up-regulating adhesion-molecule expression, facilitating additional leukocyte adherence. NK-cell cytokines act on endothelial cells increasing HLA expression with additional binding targets for DSA. Perforin and granzymes are released from NK cells, increasing endothelial damage. Activation of complement cascade leads to formation of membrane attack complex (MAC) C5b-C9, enhancing the endothelial activation [25]. Indolent microvascular abnormalities may occur without compromising acutely the liver function in transplant recipients with DSA, with the activation of stellate cells, but progressively leads to the development of chronic allograft damage, dysfunction, and eventually loss (Figure 1) [2].

The role of innate immunity in the process of post-transplant idiopathic liver fibrosis process has been featured. Toll-like receptors (TLRs) are pathogen recognition receptors that bridge the innate immunity and adaptive immune response, up-regulating cytokines and chemokines inducing dendritic cell maturation and adaptive immune activation. TLR4 enhances the chemotaxis of Kupffer cells and activation of hepatic stellate cells (HSCs), promoting liver fibrosis. TLR9 promotes the transformation of HSCs into myofibroblasts, which produce collagen fibers. TLR9 also activate Kupffer cells, the hepatic macrophages that promote fibrosis by aggregating pro-inflammatory cytokines and chemokines in early stages and secreting matrix metalloproteinases (MMPs) in late stages. Kupffer cells produce CXCL16, attracting NKT cells promoting progression of liver fibrosis [26]. There is a complex crosstalk between innate immunity and adaptive immunity in the process of fibrosis after liver transplantation.

### 3.4. Clinical and Histopathological Aspects

Recently, the clinical importance of cAMR in liver transplantation has gained attention. Table 2 shows the main pediatric studies suggesting the presence of cAMR in the long term. The real prevalence of cAMR is not clear, for indolent allograft abnormalities may occur without apparent dysfunction of liver function in DSA positive recipients, being consequently underdiagnosed. It is putatively reported to reach 8–15% of persistent or de novo DSA positive patients [2,3,5,7,8,9,10,12,14,20]. 

Miyagawa-Hayashino et al. reported the results of protocol-biopsies and their relationship to DSA for 79 pediatric patients with good liver function more than 5 years post-liver transplant (median 11 years, range 5–20 years). DSA was positive in 48% (32/67), with a higher prevalence of bridging fibrosis (88% vs. 17%, *p* < 0.001), diffuse endothelial C4d (15.6% vs. 0; *p* < 0.01), as well as mild/indeterminate acute rejection in DSA-positive patients (47% vs. 14%, *p* = 0.004), suggesting the influence of antibody-mediated rejection in the pathogenesis of liver allograft fibrosis in stable pediatric patients. This cohort included six ABO-incompatible patients (two DSA positive and four DSA negative, *p* = 0.46), as well as four tolerant patients (all DSA negative, *p* = 0.048). They showed a strong correlation of fibrosis and DSA in long-term followed up clinically stable pediatric liver-transplanted patients, suggesting the contribution of humoral immunity in the idiopathic graft fibrosis process in liver transplantation [5].

Wozniak et al, in a long-term follow-up retrospective single-center study enrolling 50 pediatric LT, compared tolerant and non-tolerant patients, showed non-tolerant patients have more class II DQ DSA positivity (61%) compared to stable (20%) or tolerant (29%) patients [6].

Feng et al., in an operational tolerance-inducing protocol for 20 pediatric patients with stable liver functions for more than 4 years, found 12 tolerant patients (60%). Interestingly, there were no statistical difference in previous DSA positive rate (4/12, 33% of tolerant and 5/6, 83% non-tolerant, *p* = ns) and de novo DSA rate (7/12, 58% of tolerant and 2/6, 33% of non-tolerant, *p* = ns), probably due to the small number of patients. Tolerant patients had less portal inflammation (91.7% [95% CI, 61.5–99.8%] vs. 42.9% [95% CI, 9.9–81.6%]; *p* = 0.04), and C4d score were lower (median of 6.1 [IQR, 5.1–9.3] vs. 12.5 [IQR, 9.3–16.8]; *p* = 0.03) compared to non-tolerant patients. This study also shows evidence of humoral influence in non-tolerant patients; however, it is not conclusive due to the small number of patients [22]. In other paper, Feng et al., in a multicenter prospective immunosuppression withdrawal study (iWITH) enrolling 157 long-term stable pediatric liver transplant patients, found that class II DSA MFI sum > 20,000 were at increased risk of higher Ishak fibrosis stage (OR, 4.53; 95% CI, 1.78–11.53, *p* = 0.001), portal inflammation grade (OR, 3.59; 95% CI, 1.30–9.93; *p* = 0.01), and C4d scores (portal capillary: OR, 5.11; 95% CI, 1.98–13.20; *p* < 0.001; sinusoidal: OR, 4.40; 95% CI, 1.49–12.98; *p* = 0.007; total: OR, 4.73; 95% CI, 1.95–11.48; *p*< 0.001). These results confirmed the influence of humoral rejection in the indolent evolution of liver allograft fibrosis in apparently stable patients [8].

Dao et al. evaluated 10-year protocol biopsies of pediatric patients and C4d and DSA of liver-transplanted ABO compatible/identical patients. All biopsies presented fibrotic changes, with a mean liver allograft fibrosis score (LAFSc) of 5.1 ± 2.2. C4d was positive in 58% (31/53) of the biopsies. DSA was positive in 48% (20/44) of the patients with a mean maximal MFI of 12,977 ± 6731. Mean LAFSc (6.3 ± 1.3 versus (vs) 3.9 ± 2.2; *p* = 0.008), perivenular fibrosis (2.7 ± 0.5 vs. 1.3 ± 1.0; *p* < 0.001), and portal inflammation (1.4 ± 0.8 vs. 0.3 ± 0.5; *p* = 0.009) were significantly higher in the double DSA-positive and C4d-positive group comparing to the double-negative group, concluding that indolent fibrosing process might be related to AMR [9].

Neves Souza L, et al. examined histopathological features of 460 liver explanted allografts at re-transplantation and found a decrease in chronic rejection rate with increased idiopathic post-transplant hepatitis (IPTH) rate along the years. In the pediatric liver retransplanted population, IPTH become the main reason of re-transplantation, accounting for 40% (8/20) of all pediatric re-transplantations from 2002 to 2014. They suggest the possibility of chronically evolving antibody-mediated rejection as cause of these fibrotic changes, but further research is needed to clarify this mechanism [27].

Evans et al., studied 158 asymptomatic children with more than 5-year graft survival submitted to protocol liver biopsies at 1, 5 and 10 years after OLT and found increasing rates of chronic hepatitis (22%, 43% and 64% at 1, 5 and 10 years, respectively) associated to allograft fibrosis (52%, 81% and 91% at 1, 5 and 10 years, respectively) along the years. They correlated these findings with the presence of autoantibodies, suggesting the influence of antibodies in the indolent hepatitis and fibrosis progression [14].

Guerra et al., in a retrospective study of 45 patients, suggested a strong correlation between obliterative portal venopathy (OPV) and presence of DSA (all 4 OPV had DSA) and C4d positivity (2 of 4 OPV presented C4d), suggesting that OPV may be a pathognomonic sign of cAMR [16].

Jackson et al., in a multicenter prospective study enrolling 129 clinically stable pediatric patients, showed that 67 (43%) presented subclinical chronic graft injury. They found that IgG4 DSA was strongly correlated with greater HLA mismatch, presence of interface activity (with variable degrees of fibrosis), and a transcriptional profile of attenuated T cell-mediated rejection [15].

Angelico et al., in a retrospective single-center study enrolling 219 pediatric patients followed-up for more than 5 years, showed a increased incidence of liver fibrosis along the time. Allograft fibrosis (AF) was present in 73% 6 months after LT, 90% in the 5^th^ year and 90% in the tenth year after LT [18].

There are some key studies suggesting the presence of cAMR based on the adult population.

Musat et al., evaluated 43 liver-transplanted ABO compatible/identical adult patients and found DSA positivity associated to C4d portal deposition in 40%, with higher acute cellular rejection (ACR) rate (88% vs. 50%, *p* = 0.02), as well as steroid-resistant ACR rate (41% vs. 19%, *p* = 0.03) in DSA positive patients. They also found DSA and C4d positivity in 6 of 10 (60%) chronic ductopenic rejection patients [11].

Sakashita reviewed the liver biopsies of 764 liver transplanted pediatric and adult patients and compared the survival rate between crossmatch (CM) negative (*n* = 749) and positive (*n* = 15) patients. 5-year survival rate was higher in the negative group (77% vs. 53%; *p* = 0.009). The C4d staining pattern in late biopsies (>90 days) was analyzed and they found that C4d was more prominent in the CM positive group (82% vs. 34%; *p* = 0.002), suggesting the possible humoral effect in liver transplantation [13].

O’Leary found DSA positivity in 92% of chronic rejection patients, suggesting a strong influence of antibody mediated rejection in these patients. They concluded that humoral rejection may be involved in the physiopathology of chronic rejection [4]. In other study, O’Leary et al. analyzed the relationship between DSA and biopsy findings in 45 DSA highly positive (MFI > 10,000) patients, comparing to 45 matched DSA negative patients with stable liver enzymes in the long term. They found a higher incidence of lobular inflammation, interface activity and peculiar patterns of fibrosis (portal tract collagenization, portal venopathy and sinusoidal fibrosis), suggesting a scoring system to evaluate chronic antibody-mediated rejection (cAMR-score). According to this score, a cutoff value of 27.5 predicted a 50% 10-year allograft failure [7].

Del Bello and colleagues found a prevalence of 13% of DSA (MFI > 1000; 35 of 267 patients), in a median of 51 months (6–220 months) post-liver transplant. 71% (25 of 35 patients) persisted with DSA in the last follow-up. They also found an incidence of de novo DSA in 9% (21 of 232 patients) and five of them developed AMR, with an incidence of 23.8% (5 of 21) among de novo DSA patients. They also found that Metavir fibrosis score was 2.14 ± 1 in patients with persistent DSA, 2.25 ± 0.9 in de novo DSA, and 0.9 ± 0.9 in those without DSA (*p* = 0.02), concluding that liver allografts of DSA-positive patients develop more fibrosis, suggesting the influence of humoral reaction in the process of liver fibrosis [10].

### 3.5. Treatment of cAMR

Treatment protocols for cAMR in liver transplantation are not established. Reports of treatment are limited to some case reports and case series [2,3,19,28].

In the practical setting, at the Kyoto University, patients diagnosed as cAMR are managed with adjustment of Tacrolimus dose, introduction of steroids and MMF. Liver biopsies are repeated more frequently. In refractory cases, mTOR inhibitors such as Everolimus are introduced. If liver dysfunction persists, IVIG, Rituximab or plasmapheresis are considered, based on ABO incompatible protocol. Retransplant is performed in refractory cases to treat liver failure.

#### 3.5.1. Calcineurin-Inhibitor (CNI) Conversion

Patients in use of Cyclosporine should be converted to Tacrolimus, as chronic rejection patients may improve their liver function with CNI conversion. For patients in IS weaning process, Tacrolimus dose should be increased or reintroduced [2,3,19,29].

#### 3.5.2. Immunosuppression Adherence

Most noncompliant patients develop repeated episodes of T-cell mediated rejection (TCMR), and some evolve to chronic rejection. Immunosuppression adherence may improve the graft function in early phases of cAMR, but it becomes irreversible in the late phase. Adolescents are especially vulnerable to non-compliance, therefore special care is needed to improve their compliance [2,3,19,24].

#### 3.5.3. mTOR Inhibitors

mTOR inhibitors, such as Everolimus and Sirolimus, inhibit cell proliferation by blocking cell cycle progression from the G1 to the S-phase by forming a complex with the immunophilin FK506-binding protein 12 (FKBP12) and inhibiting the protein kinase mTOR [30]. The association of mTOR inhibitors to CNI has been showing some improvement in chronic rejection patients. Nielsen et al. introduced Everolimus to 12 pediatric chronic liver allograft dysfunction patients evolving to normalization in four and partial improvement in six patients [31]. Ueno et al. presented two pediatric liver transplant patients who developed steroid-resistant chronic rejection and their liver function tests improved after the introduction of Everolimus without progression of fibrosis thereafter [32]. More evidence supporting the use of Everolimus in the treatment of cAMR is needed.

#### 3.5.4. Rituximab, Bortezomib and Eculizumab

Rituximab, a chimeric monoclonal IgG antibody directed to CD20 antigen present in the surface of B lymphocytes, has been utilized for the treatment of cAMR, when CNI or mTOR inhibitor do not work. Sakamoto et al., in a multicenter observational study, reported improvement of liver function in 2 of 4 cAMR pediatric patients treated with Rituximab. The two patients who had no improvement, presented severe fibrosis before the treatment [28]. However, there are no other studies supporting the use of Rituximab to treat cAMR in liver transplantation.

There is no report for Bortezomib (proteasome inhibitor) or Eculizumab (monoclonal antibody blocking complement pathway) to treat cAMR in the literature.

In refractory cAMR cases, re-transplantation is the only option for allograft failure [27]. Multicenter prospective randomized studies are needed to establish the best treatment protocol to manage cAMR in liver transplantation.

## 4. Discussion

Chronic Antibody-Mediated Rejection (cAMR) occurs in liver transplanted patients on suboptimal immunosuppression (IS). Most of the cAMR patients show normal liver function tests, with no serous markers to diagnose cAMR. DSA suggests the presence of antibodies against the liver allograft, but some patients with positive DSA have no signs of rejection in liver biopsies and some cAMR patients have no detectable DSA. Such patients may have antibodies against non-HLA minor antigens such as glutathione S-transferase T1 (GSTT1) and IgG4 DSA [15,21].

Positive C4d staining is suggestive of the humoral process, but many cAMR patients are C4d-negative. As C4d staining is not a routine procedure in all transplant centers, most cAMR remain underdiagnosed [2,5,7,13,20]. For this reason, the real incidence and prevalence of cAMR in pediatric liver transplantation remains unknown [10].

Idiopathic post-transplant hepatitis (IPTH) leading to liver fibrosis may be antibody-mediated, meaning that it can be a manifestation of cAMR, but more research is needed to confirm this relationship [17,27].

## 5. Conclusions

Chronic AMR in pediatric liver transplantation has been recognized as a new challenge for long-term survivors. However, the real incidence of this entity remains unknown since many transplantation centers do not perform protocol biopsies associated with C4d immunohistochemistry and DSA measurement. The presence of cAMR is only confirmed if it is suspected, actively performing histology, DSA serology and C4d staining.

There is no consensus in the management of cAMR. Many immunosuppression combinations have been attempted, with questionable results. Therefore, prospective multicenter randomized studies are needed to find the best treatment protocol to control cAMR.

## Figures and Tables

**Figure 1 jcm-11-04834-f001:**
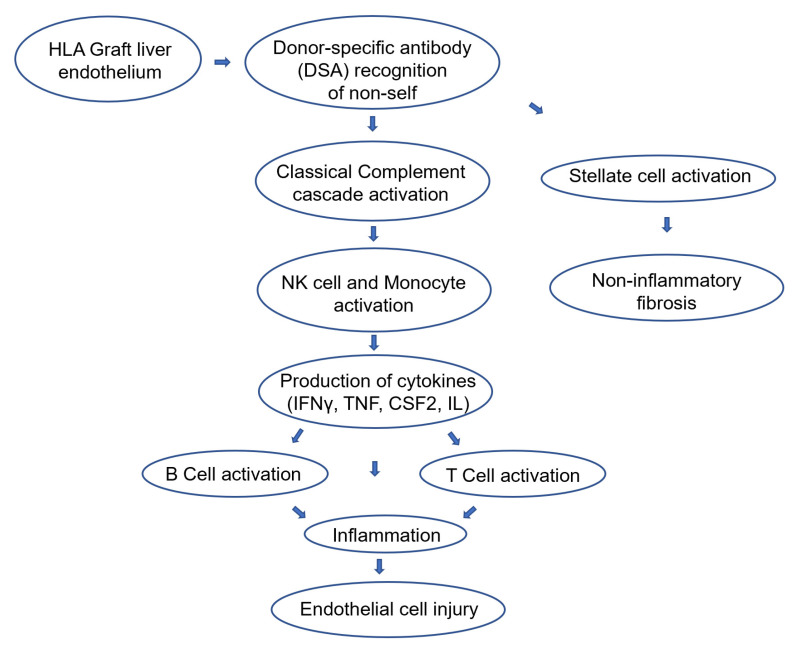
Physiopathology of AMR. HLA (Human leucocyte antigen) molecules present in the liver allograft endothelium are targeted by preformed or de novo donor-specific antibody (DSA), recognizing the graft as non-self. The classical complement cascade is activated, which subsequently activates natural-killer (NK) cells releasing interferon-γ (IFN-γ), tumor necrosis factor (TNF), and granulocyte-macrophage colony-stimulating factor (CSF2), and monocytes releasing TNF, interleukin-1 (IL-1), and interleukin-6 (IL-6), evoking endothelial injury. These cytokines also activate acquired immunity, B cells leading to AMR as well as T cells leading to T cell-mediated rejection (TCMR). DSA also activate Stellate cells, leading to liver fibrosis.

**Table 1 jcm-11-04834-t001:** Chronic antibody-mediated rejection characteristics.

Chronic Antibody-Mediated Rejection (cAMR)	
Histological findings	Probable cAMR (all four criteria are required): (1) (a) Otherwise unexplained and at least mild mononuclear portal and/or perivenular inflammation with interface and/or perivenular necro-inflammatory activity; AND (b) At least moderate portal/periportal, sinusoidal and/or perivenular fibrosis; AND (2) At least focal C4d staining in >10% of the portal tracts microvascular endothelia; AND (3) Circulating DSAs in serum samples collected within 3 months of biopsy; AND (4) Other causes have reasonably been excluded.
	Possible cAMR: (1) As above, but C4d staining is minimal or absent
Incidence	Unknown; estimated to be present in 8 to 15% of de novo or persistent DSA
Risk factors	(1) Presence of DSA (especially de novo anti-HLA class II antigens). (2) Inadequate IS (cyclosporine-based regimens or low CNI concentrations, low compliance). (3) MELD score > 15. (4) Young age at transplantation. (5) Re-transplantation. (6) allograft fibrosis. (7) IgG3 and C1q+ DSA. (8) GSTT1positive donor to negative recipient. (9) Angiotensin II type 1 receptor antibody positive recipient.
Clinical implications	Increased fibrosis and graft failure in an unknown percentage of patients

Abbreviations: cAMR: chronic Antibody-mediated rejection; DSA: Donor-specific antibody; IS: immuno-suppression; CNI: calcineurin inhibitors; MELD Mayo End-Stage Liver Disease; HLA: Human Leucocyte Antigen; GSTT1: Glutathione S-Transferase Theta1.

**Table 2 jcm-11-04834-t002:** Studies of long-term followed up liver transplanted pediatric patients suggesting the presence of antibody-mediated chronic rejection.

Authors	Year	Number of Patients	Time after LT	DSA+	C4d	Histological Findings	Type of Study
Miyagawa-Hayashino A, et al. [5]	2012	79	median 11 (5–20) years	48% (32/67)	15.6% of DSA+	DSA+: present more bridging fibrosis, endothelial C4d, acute rejection	Single center, retrospective
Wozniak L, et al. [6]	2015	50	3.7 ± 4.4 years at LT; 16 ± 4.9 years at study	54%	N.A.	Non-tolerant patients have more DQ DSA positivity (61%) compared to stable (20%) or tolerant (29%) patients.	Single center, retrospective
Feng S, et al. [8]	2018	157	8.9 ± 3.46 years	Class II 55.6% (80/144)	Score 0–3: 29%; 4–6: 42%; 7–9: 18%; >9: 10%	DSA class II+: more fibrosis, portal inflammation and higher C4d score	Multicenter, prospective
Dao M, et al. [9]	2018	53	131.3 ± 15.3 months	48% (20/44)	48% (31/53)	LAFSc, perivenular fibrosis and portal inflammation higher in double DSA and C4d positive	Single center, retrospective
Neves Souza L, et al. [27]	2018	118 pediatric retransplants	>10 years post LT	N.A.	N.A.	Increased incidence of IPTH among children (40%) in the recent era	Single center, retrospective
Evans HM, et al. [14]	2006	158	>5 years post LT (protocol biopsies 1, 5, and 10 years after LT)	N.A.	N.A.	Increasing rates of chronic hepatitis (22%, 43%, 64% at 1, 5, 10 years post LT) and allograft fibrosis (52%, 81%, 91%) along the years	Single center, retrospective
Guerra MAR, et al. [16]	2018	45	2–14 years post LT	Positive in all 4 patients with OPV	Positive in 2 of 4 patients with OPV	OPV was present in four patients with cAMR features	Single center, retrospective
Jackson AM, et al. [15]	2020	129	1.9 (1.74) at LT, 10.9 (3.54) at study	65 (50%)	N.A.	67 (43%) subclinical chronic graft injury	Multicenter, prospective
Angelico R, et al. [18]	2022	80	>5 years	N.A.	N.A.	AF 6 mo after LT: 73.8% AF 5 y after LT: 90% AF 10 y after LT: 90% Risk factors for AF: CIT > 8 h, donor ager > 40 y, low FK trough 1 y post LT.	Single center, retrospective

Abbreviations: LT, liver transplantation; DSA, donor-specific antibody; HLA, human-leukocyte antigen; N.A., not available; LAFSc, liver allograft fibrosis score; IPTH, Idiopathic post-transplant hepatitis; OPV, obliterative portal venopathy; cAMR, chronic antibody-mediated rejection; mo, months; y, years; AF, allograft fibrosis; CIT, cold ischemic time.

## Data Availability

All data obtained for this study are available from the corresponding author (EYU) upon reasonable request.

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
