# Peer review of "The New Challenge in Pediatric Liver Transplantation: Chronic Antibody-Mediated Rejection"

_jcm, 2022, doi:10.3390/jcm11164834_

Round 1
Reviewer 1 Report
I read with interest Uebayashi et all. article.
However, some points need to be addressed:
- The type of review: systems? Descriptive? If systematic, need the PRISMA
- The focus on the pediatric population is not clear and specific on the text, mixed with different information not related to the pediatric population. A table could be helpful for the reader.
- A recent article published by Lee et al. (doi.org/10.1016/j.jhep.2021.07.027) is very comprehensive, and this article didn't seem to add more information.
Thank you
Author Response
Thank you for your precious comments.
This review is a descriptive review.
I agree with you. There are few articles about chronic AMR centered on the pediatric population. Therefore, I needed to include some articles based on the adult population to illustrate some aspects of cAMR.
Thank you for the advice to create a table about the articles centered on pediatric patients to make it easier to understand.
I agree with you, Lee, et al wrote a great review including pediatric and adult populations about a similar topic.
Thank you for your precious advice to improve this article. I hope we can answer your expectations.
Sincerely yours,
Elena Yukie Uebayashi.
Reviewer 2 Report
This is a well written review on a difficult topic - cAMR. Table 1 is not formatted appropriately and it is difficult to understand. It can be improved, maybe dividing in 2 tables. Also, the author could think of including a figure or diagram depicting the mechanisms involved in the physiopathology of cAMR. Also, it would be interesting selecting some variables in order to show the available publications in the pediatric age range. Is there more literature concerning long term outcomes of ABO incompatible LT and cAMR, and the existing desensitisation protocols?
Author Response
Thank you for your precious comments. Yes, cAMR is an extremely difficult topic. According to your suggestion, I improved the table. I included a figure to illustrate the mechanisms involved in the physiopathology of cAMR. Based on your suggestion, I made a table with the publications available in the pediatric population. Actually, I found no specific literature regarding cAMR in long-term ABO-incompatible liver transplantations, for most of the papers exclude ABO-incompatible patients, probably to avoid any bias. The only paper including ABO-incompatible was written by Miyagawa-Hayashino, et al. It included six ABO-incompatible patients in their cohort, and found no difference in the DSA positivity. Thank you for your advice on improving this article.Sincerely yours,
Elena.
Round 2
Reviewer 1 Report
I read with interest the article of Uebayashi et al. It is a comprehensive review about Chronic 2 Antibody-Mediated Rejection in the pediatric population.
Few points:
- Methods: How the paper search/review has been performed? do you use other inclusion/exclusion criteria (age, date of publication..)?
- in the physiopathology section would be great to have a focus on the pediatric population. Is there any difference between this population compared to the adult one? are most of the data from the adult or pediatric populations?
- same in the treatment section. Any difference in treatment between the adult and pediatric populations?
Thank you
Author Response
Thank you for the interesting comments.
About the review method, I tried to focus on articles written after 2010. However, I needed to cite some older articles, such as about the efficacy of tacrolimus switching and CNI adherence, as well as articles about the pharmacological mechanism of Tacrolimus and Everolimus. The majority of the articles are centered on the pediatric population, but I needed to cite some articles on the adult population about the diagnostic criteria of chronic AMR.
The pediatric population has fewer recurrences after LT (such as viral hepatitis) compared to the adult population, leading to fewer biases to analyze the relationship between DSA and liver fibrosis to study cAMR. Most of the data about chronic AMR are based on long-term followed-up patients with protocol biopsies associated with DSA and C4d evaluation in the pediatric population.
As cited in the review, the pediatric population (especially adolescents) is vulnerable to decreased adherence (CNI, steroids), which may facilitate the development of de novo DSA, increasing the chance to develop cAMR.
Thank you again for the precious comments to improve this paper.
Sincerely,
Elena Yukie Uebayashi